# Negative Social Tipping Dynamics Resulting from and Reinforcing Earth System Destabilisation

Viktoria Spaiser[1], Sirkku Juhola[2], Sara M. Constantino[3], Weisi Guo[4], Tabitha Watson[5], Jana Sillmann[6], Alessandro Craparo[7], Ashleigh Basel[8], John T. Bruun[9], Krishna Krishnamurthy[10], Jürgen Scheffran[11], Patricia Pinho[12], Uche T. Okpara[13], Jonathan F. Donges[14,22], Avit Bhowmik[15], Taha Yasseri[16,23], Ricardo Safra de Campos[5], Graeme S. Cumming[17], Hugues Chenet[18], Florian Krampe[19], Jesse F. Abrams[5], James G. Dyke[5], Stefanie Rynders[20], Yevgeny Aksenov[20], Bryan M. Spears[21]

[1] School of Politics and International Studies, University of Leeds, Leeds, LS2 9JT, United Kingdom
[2] Department of Environmental Sciences, University of Helsinki, Helsinki, 00790, Finland
[3] Doerr School of Sustainability, Stanford University, Palo Alto, CA, 94305, United States
[4] Centre for Autonomous and Cyberphysical Systems, Cranfield University, London, MK43 0AL, United Kingdom
[5] Global Systems Institute, University of Exeter, Exeter, EX4 4QE, United Kingdom
[6] Cluster of Excellence Climate, Climatic Change, and Society, Hamburg University, Hamburg, 20146, Germany
[7] International Centre for Tropical Agriculture (CIAT), Recta Cali-Palmira, Valle del Cauca, Colombia
[8] Alliance Biodiversity-CIAT, Cape Town, 7600, South Africa
[9] Faculty of Environment, Science and Economy, University of Exeter, Exeter, EX4 4QE, United Kingdom
[10] Meru Labs, Panama City, 0700, Panama
[11] Institute of Geography, Research Group Climate Change and Security, Hamburg University, Hamburg, 20144, Germany
[12] Amazon Environmental Research Institute, Altamira, 68373-100, Brazil
[13] Natural Resources Institute, University of Greenwich, Kent, ME4 4TB, United Kingdom
[14] Earth Resilience Science Unit, Potsdam Institute for Climate Impact Research, Member of the Leibniz Association, Potsdam, 14473, Germany
[15] Risk and Environmental Studies, Karlstad University, Karlstad, 65188, Sweden
[16] School of Sociology, University College Dublin, Dublin, 8Q4G 8Q, Ireland
[17] Oceans Institute, University of Western Australia, Perth WA 6009, Australia
[18] IESEG School of Management, Univ. Lille, CNRS, UMR 9221 - LEM - Lille Economie Management, F-59000 Lille, France
[19] Stockholm International Peace Research Institute, Stockholm, 169 72, Sweden
[20] National Oceanography Centre, Southampton, SO14 3ZH, United Kingdom
[21] UK Centre for Ecology & Hydrology, Edinburgh, EH26 0QB, United Kingdom
[22] Stockholm Resilience Centre, Stockholm University, Stockholm, 10691, Sweden
[23] Geary Institute for Public Policy, University College Dublin, Dublin, D04 N9Y1, Ireland

*Correspondence to*: Viktoria Spaiser (v.spaiser@leeds.ac.uk)

**Abstract.** In recent years research on normatively positive social tipping dynamics in response to the climate crisis has produced invaluable insights. In contrast, relatively little attention has been given to the potentially negative social tipping processes that might unfold due to an increasingly destabilised Earth system, and how they might in turn reinforce social and ecological destabilisation dynamics and/or impede positive social change. In this paper, we discuss selected potential negative social tipping processes (anomie, radicalisation and polarisation, displacement, conflict and financial destabilisation), linked to Earth system destabilisation. We draw on related research to understand the drivers and likelihood of these negative tipping dynamics, their potential effects on human societies and the Earth system, and the potential for cascading interactions (e.g. food insecurity and displacement), contributing to systemic risks. This first attempt to provide an explorative conceptualisation and empirical account of potential negative social tipping dynamics linked to Earth system destabilisation is intended to motivate further research into an under-studied area that is nonetheless crucial for our ability to respond to the climate crisis and for ensuring that positive social tipping dynamics are not averted by negative ones.

## 1 Introduction

Recent advances in research on Earth system tipping points (ESTPs) (e.g. Armstrong McKay et al., 2022), paint an increasingly alarming picture of the state of our planetary system. Understanding tipping points and other forms of non-linear change is now widely recognised as critical to managing and responding to change in complex systems (Scheffer, 2009). We define

social tipping points on the basis of mathematics of dynamical systems (Strogatz, 2000). Specifically, tipping points in dynamical social systems are critical thresholds where a small change in a variable describing the state of the social system or in a parameter capturing external influences leads to an often abrupt qualitative change of the dynamical social system, i.e. the social dynamical system undergoes a phase transition from one state to another (Winkelmann et al 2022). Tipping occurs because positive feedback mechanisms create self-reinforcing loops, where a small change in one component of the system triggers changes that further reinforce the initial change. Tipping is further enabled by weak negative feedback mechanisms that tend to stabilise a dynamical system. Tipping is usually difficult to reverse due to hysteresis that locks the system within the new state or within the trajectory to a new state, even if the original drivers for the change are removed (Wiedermann et al., 2020, Winkelmann et al., 2022). Normatively speaking, social tipping points can be both positive (predominantly beneficial to humans and the natural systems) or negative if they result in catastrophic consequences for human societies and ecological systems (IPCC, 2022; Lenton et al., 2023).

Increasing attention is also being paid to cascade effects that connect different systems, implying that a change in one system may trigger further change in another system (Liu et al., 2023). Here, we consider a tipping cascade to take place when one tipping point triggers the crossing of another tipping point (Klose et al., 2021). We focus here moreover on negative social tipping processes that have the potential to feed back to the Earth system, further destabilising it, i.e. we are interested in processes where the Earth system destabilisation contributes to social system destabilisation, which then further destabilises the Earth system (e.g. due to lack of cooperation), creating a potential feedback loop. We note that this paper focuses on climate ESTPs, but the same rationale can be broadly generalised to other Earth systems.

Although research on the potential for positive social tipping dynamics in various systems (e.g. food, energy, transportation, financial, behavioural etc.) has started to emerge (Tàbara et al., 2018; Otto et al., 2020; Lenton 2020; Lenton et al., 2022; Winkelmann et al., 2022; Milkoreit, 2023), there has been limited research on negative social tipping dynamics that might be triggered by climate change (Laybourn et al., 2023). This is noteworthy, not least because early research on tipping points in the social sciences was mostly concerned with undesirable social processes, such as rapid and non-linear patterns of urban racial segregation in the United States (Schelling, 1978). More recently, researchers have used dynamical systems analyses to empirically study tipping points in school segregation (Spaiser et al., 2018), political instability of countries (Grimm & Schneider, 2011), and rapid proliferation of misinformation (Törnberg, 2018).

We argue that studying negative social tipping points in the context of Earth system destabilisation is important because it highlights the risks generated by overshooting temperature thresholds such as 1.5°C (Bustamante et al., 2024). While indeed every tenth of a degree of temperature increase matters, framing around climate policy is moving in the direction of making overshoot socially acceptable. Overshoots are presented as temporary, with the deployment of carbon dioxide removal (CDR) being able to recover temperatures back into the 'safe zone' by the end of the century. The risks of ESTPs however make overshooting very dangerous, as overshooting may trigger ESTPs, which then cannot be reversed even if we return to lower global warming after the overshoot period. Triggering ESTPs on the other hand poses the risk of escalating climate change impacts (Wunderling et al., 2024). Moreover, overshooting would lead to further ecological destabilisation (Singh et al., 2023), which might be reversible in terms of returning to lower global warming; but in the meantime, ecological destabilisation could trigger negative social tipping points described here, and these negative social tipping points could feedback to the Earth system, further destabilising it, potentially leading to ESTPs being triggered. We believe that understanding these potential complex interactions is important, because humans have agency and can make decisions trying to prevent such escalating processes. None of the scenarios described here is inevitable and although many dynamics are already unfolding today, we have not reached a point of no return.

Negative and potentially catastrophic consequences are unequally distributed, both internationally and within individual
societies (Pereira et al., 2024). Research has emphasised that low-income countries that have often contributed least to the
destabilisation of the Earth system will bear the brunt of the climate change impacts (IPCC, 2022; Lenton et al., 2023).
Moreover, within each society, it is the most vulnerable groups, such as children (Thiery et al., 2021; UNICEF, 2021), women
(Denton, 2002), minority groups (Berberian et al., 2022, Donaghy et al., 2023) and generally the less affluent (Thomas et al.,
2019), who will be most affected by climate change impacts. Triggering negative social tipping points will have considerable
consequences for these vulnerable groups, further amplifying their vulnerability and stressing the need for climate justice
(Newell et al., 2021).
In this perspective, we pose the following questions: (1) What are the potential negative social tipping points that the
destabilisation of the Earth system could trigger? (2) To what extent could the triggering of negative social tipping points
further destabilise the Earth system? (3) How do these negative tipping elements interact and what cascades could these
interactions cause? (4) What research and modelling approaches are suitable for studying negative social tipping points and
cascades? And (5) what intervention options are available to prevent negative social tipping points and cascades?

## 2 Mapping out Negative Social Tipping

We identify five negative social tipping processes that according to some existing evidence could be triggered by Earth system
destabilisation (see Figure 1). The part or subsystem of a larger system that can pass a tipping point is referred to as the tipping
element. Drawing on the positive social tipping element framework developed by Otto et al. (2020), we identify four social
tipping elements (TE) that have the potential for negative tipping processes (TP): socio-psychological systems (TE1), political
systems (TE2), human settlements (TE3) and financial markets (TE4). Figure 1 provides an overview of these tipping elements
and the tipping that could be triggered within these tipping elements: Anomie (TP1.1), Radicalisation & Polarisation
(TP1.2), Displacement (TP2.1), Conflict (TP3.1) and Financial Destabilisation (TP4.1). All these processes can unfold across
different levels of social structure on different time- and spatial scales. Specifically, tipping can occur as rapidly as hours,
triggered by a major shock event or unfold more slowly (years) over cascading pathways as the effects of ESTPs accumulate.
Tipping can also occur only locally, affecting a specific community or spread across a nation or the globe. The figure also
indicates the potential for interactions between various negative tipping elements. The interactions between different TEs
indicate different possible destabilisation pathways that could lead to the crossing of negative tipping points across scales. This
illustrative selection is based on evidence for tipping processes in these subsystems and evidence that Earth system
destabilisation has a direct effect on these subsystems.
[FIGURE 1 HERE]

### 2.1 Anomie

The concept of anomie, which was introduced by Durkheim (1893, 1897) to describe the breakdown of norms and social order
and its relationship to suicide patterns in societies, has evolved over decades of social research (Abrutyn, 2019; Twyman-
Ghoshal, 2021). We define anomie as a state of a society or community that is characterised by a breakdown of social norms,
social trust, social ties, and social reality, resulting in social disorder and disorganisation, disorientation, and disconnection.
These syndromes manifest on the individual level through mental health deterioration, increased suicide rates, and/or increased
deviant behaviour (Brown, 2022; Teymoori et al., 2017). Although this is a relatively new area of research, there is increasing
evidence to suggest that changes in the Earth system can contribute to anomie. For instance, anomie has been observed in the
aftermath of natural disasters, made more likely by climate change (Miller, 2016) and it has been suggested that Earth system
destabilisation may result in a new form of anomie, called environmental anomie (Brown, 2022), where sudden changes to the

physical landscape can upend the established social order and undermine people's ability to comprehend, relate to, and function within their environment. For instance, people from Paradise (California, US) who survived the devastating Camp Fire in 2018 reported how the wildfire event undermined their ability to comprehend the world around them, because their familiar environment became unintelligible (e.g. they struggled to determine wind direction), they were no longer able to relate to and function within their environment. This resulted in a breakdown of self-efficacy, with a sense of unreality taking hold (e.g. burning tree branches falling from the sky). This experience of environmental anomie was further exacerbated when the affected individuals witnessed that traditional authorities were overwhelmed and unable to respond to the physical chaos, which undermined confidence and led to an individuation of suffering and feelings of social isolation, i.e. experience of general anomie. With the breakdown of social order people were forced to fend for themselves and rules (e.g. regulating traffic) were no longer observed (Brown, 2022).

Beyond anomie resulting from extreme weather events caused by escalating climate change, there is also evidence for a rise in anomic experiences, particularly by young people and children around the world, contributing to a mental health crisis. In a first comprehensive study, surveying 10,000 children and young people (aged 16-25 years) in 10 countries (Australia, Brazil, Finland, France, India, Nigeria, Philippines, Portugal, UK and USA) Hickman et al. (2021) found that more than 45% said their feelings about climate change negatively affected their daily life and functioning, 75% reported that they find the future frightening, and 83% said they think people (adults) have failed to take care of the planet. But it is not just the young experiencing the effects of climate change on mental health – it is negatively affecting the mental health and emotional wellbeing of people of all ages globally, but more profoundly of poor and vulnerable populations (Lawrence et al., 2021; Clayton et al., 2017), as well as women and Indigenous people (IPCC, 2022; Sultana, 2022). For a summary of other studies, see Figure 2.

[FIGURE 2 HERE]

The extent of tipping dynamics in anomie have not yet been studied directly, but some studies have demonstrated tipping dynamics in phenomena that can serve as proxies for the anomic state of a society or community. Specifically, (complex) contagion processes (see Table 1) have been observed for mental disorders and distress, including suicide (Scatà et al., 2018; Paz, 2022), for deviant behaviours (Busching and Krahé, 2018), and for distrust (Ross et al., 2022). Societies or communities that are already in a zone of social instability (e.g. high rates of anti-social behaviour, increasing deviant behaviour such as crime or substance abuse, high rates of mental health problems) due to other factors, such as poverty, rising inequality and failing institution (Burns, 2015) or because of a gradual erosion of social norms that can affect affluent communities too (Pfiff et al., 2012; Bursztyn et al., 2020), are particular at risk to tip into an anomic state, when additionally being faced with ecological destabilisation (cf. Douglas et al., 2016). Anomie tipping can also result from a single extreme event, for instance, triggered by an ESTP being breached. Such an event can instantly disintegrate whole communities, scattering members of the community in the aftermath (i.e., interaction with displacement), leaving them with depleted social and mental resources (Miller, 2016) and establishing the perception that society as a whole is failing (Teymoori et al., 2017). Tipping in this case can be described using Logistic Map models (Bruun et al., 2017), which can model how coupled systems can tumble towards chaotic system behaviour (see Table 1). Natural and human-caused disasters can bring communities together and strengthen cooperation, however research suggests that when the experience of solidarity and unity in the disaster aftermath starts to wane, communities can experience increasing disillusionment and depression, followed by social disintegration (i.e. anomie), if they are left without adequate, long-term support (Townshend et al., 2015).

Anomie can have feedback effects on the Earth system, further destabilising it through various pathways. When social norms disintegrate, certain pro-social behaviours and collective actions that are necessary to slow down the climate crisis may

diminish (Constantino et al., 2022; Schneider and van der Linden, 2023; Lettinga et al., 2020). Without strong social norms and social ties supporting collective action and fostering reciprocity, trust, and cooperation, it becomes increasingly challenging to implement effective measures to address accelerating Earth system destabilisation, hence increasing the likelihood for Earth system tipping (Fehr et al., 2002; Thøgersen, 2008; Malerba, 2022). Moreover, mental health problems weaken people's capacity to seek solutions, fostering collective inertia and increasing susceptibility to conspiracy theories, potentially further undermining trust and cooperation to prevent further Earth destabilisation (Burden et al, 2017; de la Sablonnière & Taylor 2020; Green et al., 2023).

**2.2 Radicalisation & Polarization**

Radicalisation can be a reaction to perceived external threats, including ecological threats. Research suggests that people can respond to climate change and other ecological threats by becoming more authoritarian and derogative against outgroups (Fritsche et al., 2012; Jackson et al., 2019; Taylor, 2019; Russo et al., 2020; Uenal et al., 2021; Spaiser et al., 2024). This effect can be further exacerbated by the well documented effect of heat on aggressive behaviours, including online hate speech (Stechemesser et al., 2022). Current trends seem to suggest increasing polarisation (Dunlap et al., 2016; Vihma et al., 2021; Cole et al., 2023; Smith et al., 2024), i.e. a rise of the political right, which is increasingly attracting the political centre (Levitsky and Ziblatt, 2018; Halikiopoulou, 2018; Layton et al. 2021), obstructing climate action and increasingly diverging from the political left/centre-left, which is demanding climate action (Aasen, 2017; Lockwood, 2018; Gustafson et al., 2019). This polarisation is driven indirectly by Earth destabilisation too, as it is at least partly a response to climate mitigation policies that are perceived as a threat to the existing socio-economic system, status and identity (Dunlap et al., 2016; Hoffarth and Hodson, 2016; Dagett, 2018; Clarke et al., 2019; Benegal and Homan, 2021; Ehret et al., 2022; Brännlund and Peterson, 2024) and can be further exacerbated by inequality and general economic decline (Winkler, 2019; Stewart et al., 2020; Hübscher et al., 2023), which again can be partly linked to Earth destabilisation at least in some parts of the world (Méjean et al., 2024; Dietz et al, 2021). However, as climate change progresses and becomes a more concrete existential threat throughout the world (Huggel et al., 2022), we may see even socially liberal individuals developing increasingly authoritarian and reactionary views (Gadarian, 2010; Hetherington and Suhay, 2011; Huddy and Feldmann, 2011; Hirsch, 2022). At that stage we may see radicalisation taking a different direction, with currently fringe political ideologies such as ecofascism taking hold. Ecofascism reinterprets white supremacy ideology in the context of climate/ecological crisis with the goal to defend habitable areas for the white race and decrease world population (Taylor, 2019). Already, a couple of recent right-wing terrorists have self-identified as ecofascists, such as Brenton Tarrant, who killed 51 people during a terror attack on a mosque in Christchurch, New Zealand in 2019. A few months later Patrick Wood Crusius killed 23 people in El Paso, United States, legitimising his actions again with ecofascist ideologies (Achenbach, 2019). Certain ecofascist themes seem to also appear increasingly in public debates (Thomas and Gosink, 2021).

Radicalisation can exhibit tipping dynamics. Research has described radicalisation, e.g., the spread of right-wing ideology (Youngblood, 2020), through complex contagion processes (see Table 1). Similarly, the spreading of extremist content on social media has been observed to follow complex contagion processes (Ferrara, 2017). Indeed, polarisation and radicalisation around climate change has been observed to be on the rise online (Weber et al., 2020; Teen et al., 2020; Falkenberg et al. 2022), at times displaying non-linear, accelerating diffusion dynamics (Centre for Countering Digital Hate, 2023) and fuelled by corporate funding (Farrell, 2016; Teen et al., 2020). Moreover, processes of "cross-pollination", the merging or previously separate radical clusters facilitating further contagion, have been documented (Kimmel, 2018; Baele et al., 2023), including for climate denial (Agius et al., 2020). Polarization has also been observed to follow tipping dynamics. Leonard et al. (2021) describe for instance for the US how subtle public opinion shifts from left and right can have a differential effect on the self-reinforcing processes of elites, causing Republicans to polarize more quickly than Democrats. As self-reinforcement pushes

societies toward the critical threshold, polarisation speeds up. Political polarisation tipping, often accompanying radicalisation of certain segments of the population, has been found to be difficult to reverse due to asymmetric self-perpetuating trajectories (Macy et al., 2021).

Radicalisation and polarisation can have feedback effects on the Earth's system, destabilising it further. According to research (Stanley et al., 2017; Stanley and Wilson, 2019; Julhä and Hellmer, 2020), authoritarian and social dominance attitudes are negatively related to environmental attitudes and support for environmental/climate change policies. Indeed, right-wing ideology has been repeatedly correlated with climate change denial (Hornsey et.al, 2016; Hoffarth and Hodson, 2016; Czarnek et al., 2020; Julhä and Hellmer, 2020). When climate change is denied, no attempts are made to mitigate climate change, on the contrary, decisions may be taken to further prop up high-emitting industries (Ekberg et al., 2023; Darian-Smith, 2023). There is however increasingly a retreat of pure climate denial (primary climate obstruction), instead we see a rise in secondary and tertiary climate obstruction, which can include deliberate polarisation of societies on the issue (Kousser and Trantr, 2018; Goldberg and Vandenberg, 2019; Lamb et al., 2020; Mann, 2021; Flores et al., 2022; Ekberg et al., 2023; Burgess et al., 2024). Research moreover demonstrates that the increasing success of the radical right influences also the policies of mainstream parties (Abou-Chadi and Krause, 2020), i.e. even if radical parties are not in government, they still can undermine climate policies.

## 2.3 Displacement

Acute and slow-onset environmental pressures, such as heatwaves, long-term temperature and humidity changes, extreme weather events and sea level rise (e.g. due to the melting of Greenland glaciers, and the West Antarctic Ice Sheet), are likely to impact the migration (voluntary) and displacement (forced, involuntary) circumstances of a large proportion of the global population (Mastorillo et al., 2016; Berlemann et al., 2020; Hauer et al., 2020; Hoffmann et al., 2020; Lu and Romps, 2023). In the context of ESTPs, sea-level rise is projected to be one of the most costly and irreversible consequences of climate change (Hauer et al., 2020, McLeman, 2018, Kaczan & Orgill-Meyer, 2020; Armstrong McKay et al., 2022). Another rapid-onset hazard is land degradation due to permafrost melt, both in coastal areas and inland (Irrgang et al., 2022; Streletskiy et al., 2023). Accelerated Polar warming or Arctic Amplification warms Arctic surface temperatures by a factor two-to-four times faster than the rest of the globe (Rantanen et al., 2022), which - in addition to the direct impact on permafrost thawing - results in the loss of protective sea ice and, consequently, rapidly increasing coastal erosion (Casas-Prat and Wang 2020; Nielsen et al., 2022; Wunderling et al., 2024). As the proportion of the global population living in coastal regions continues to grow, likely surpassing one billion people this century, this will have profound implications for both individuals and societies (Hauer et al., 2020, McLeman, 2018, Kaczan and Orgill-Meyer, 2020). However, sea level rise is not the only driver of adaptive mobility (Gioli et al., 2016). Even if international efforts towards mitigating climate change are successful (RCP 4.5 – low emissions scenario), models have projected drought-induced international displacement to increase substantially by the end of the 21st Century. High emissions scenarios (e.g. RCP 8.5) would push the number of displaced due to droughts even further up (Smirnov et al., 2023).

Displacement can happen suddenly and amplifying or positive feedbacks can increase or maintain the dislocation of populations even after the extreme weather event or initial shock has passed. This can create a cycle that reinforces, extends, or renders the displacement permanent. Displaced populations must grapple with the loss of their livelihoods, often by identifying new temporary sources of income that can become permanent due to the challenges of returning to origin communities (Young and Jacobsen, 2013; Wilson, 2020). The displacement is often linked with turning away from traditional ways of life and economical support, e.g. in the cases of Arctic Inuit population fishing, hunting, and trapping (Ford et al., 2023; Streletskiy et al., 2023), and the movement away from traditional agricultural and pastoralist livelihoods in areas of Central and Southwest Africa (Akinbami, 2021; Thorn et al., 2023). This can result in cultural heritage loss (Pearson, 2023).

These compounding and reinforcing effects can exacerbate pre-existing social inequities and determine the pattern of displacement (e.g. short or long-term/permanent) among different populations (Lama, 2021; Boas et al, 2022). Additionally, with slow-onset events, decisions to migrate can be driven by social networks and connections; when members of a community migrate, others may make the decision to follow (Manchin and Orazbayev, 2018; Thorn et al., 2023; Tubi and Israeli, 2023). This can, in and of itself, be subject to tipping dynamics; when a certain percentage of a community has left, this has been observed to negatively impact those left behind, potentially triggering subsequent outmigration (Rai, 2022).

In the absence of appropriate governance mechanisms and protocols for how and where to relocate displaced communities, negative feedback consequences for the Earth systems are possible (Islam et al., 2021; Thorn et al., 2023). Hosting communities may face strains on their natural resources and/or sinks to meet the additional needs of the displaced. For example, Tafere (2018) identified environmental degradation resulting from the influx of displaced populations in East Africa, often in environmentally sensitive (e.g. protected forests) or already strained regions (e.g. arid or semi-arid areas). Such straining of ecological systems to accommodate increased ecoservices demand due to forced migration could contribute to accelerating at the very least regional ecological destabilisation.

**2.4 Conflict**

Despite growing concerns about conflict, the causal link between climate change and conflicts as well as their underlying dynamics remain debated (Burke et al., 2009; Buhaug, 2010; Buhaug et al., 2014; Solow, 2013, Kelley et al., 2015; Selby et al., 2017). While statistical models inferred either significant coincidences of particular civil conflict events with concurrent climate extreme events or significant associations of warming and drought trends with civil conflict trends, many qualitative in-depth assessments of the particular civil conflict events and their underlying mechanisms dismiss such coincidences and associations (e.g. Buhaug, 2010; Selby et al., 2017). Though not the only cause (Sakaguchi et al., 2017; Mach et al., 2019; Scartozzi, 2020; Ge et al., 2022), climate change undermines human livelihoods and security, because it increases the vulnerability of populations (e.g. to extreme events, food/water scarcity), grievances, and political tensions through an array of indirect – at times non-linear and latent (i.e. not measurable) – pathways, thereby increasing human insecurity and the risk of violent conflict (Scheffran et al., 2012; van Baalen and Mobjörk, 2017; Koubi, 2019; von Uexkull and Buhaug, 2021; Ide et al., 2023). It is difficult to separate mutually enforcing vulnerabilities to both climate and conflict that trigger an escalating spiral of violence and amplify cascading crisis events beyond critical thresholds (Buhaug and von Uexkull, 2021) and connected through telecoupling (Franzke et al., 2022).

Many conflicts can be described in terms of social tipping mechanisms, which can be triggered by Earth system destabilisation, where causal mechanisms are inferred using data (Sun et al., 2022) and can be modelled through socially connected tipping dynamics, for instance using the logistic map approach (see Table 1) (Guo et al., 2018, Aquino et al., 2019, Ge et al., 2022, Guo et al., 2023). Using a complex systems lens and connecting the human–environmental–climate security (HECS) nexus framework (Daoudy, 2021; Daoudy et al., 2022; Scheffran et al., 2012) and the social feedback loop (SFL) framework (Kolmes, 2008) can help clarify conflict tipping mechanisms in coupled social-ecological systems. The HECS framework infers that climatic drivers of civil conflicts are best understood as a result of policy decisions and governance that reflect the ideology and preferences of ruling elites or ethnic bias instead of investigating the direct functions of climate extremes. SFL suggests that initial social disruptions directly caused by gradual climate change and climate extreme events can itself generate a distinct positive feedback loop leading to self-accelerating rates of societal disintegration and to civil conflicts (Kolmes, 2008). In turn, using a combined HECS-SFL lens, civil conflicts can be perceived as amplified social disintegration and disruption resulting from societal and political responses to the initial disintegration and disruptions caused directly by climate extremes and climate change (Scheffran et al., 2023). Self-reinforcing feedbacks emerge in social-ecological systems as a result of complex interactions among socio-economic, environmental and political events and variables, such as institutional

capacity for solving social-ecological problems initially caused by climate change (Daoudy et al., 2022). These complex interactions result in the amplification of social-ecological shocks that climate change and extremes initially caused and potentially disrupt and negatively tip the system in concern to a conflict state. The affected system becomes entrapped in the conflict state until sufficient incentives can move it out. However, there remain gaps in understanding latent mechanisms which introduce variable delay (e.g. slow social transformations), confounding factors, non-linear bifurcations (e.g. some transformations are irreversible) and regional variability.

When conflicts escalate, exhibiting a tipping dynamic, they can in turn impact the Earth system, either directly as warfare itself is producing excessive GHG emissions and destroying vital ecosystems such as forests, as is for instance currently the case of Russia's war in Ukraine (de Klerk et al., 2022). For example, the Kakhovka Dam was destroyed in 2023 during the Russia-Ukraine conflict. Early assessments (UKCEH & HRW, 2023; UNEP, 2023) indicated a maximum downstream flood extent of around 83,000 hectares (6 - 9 March 2023) including inundation of downstream urban areas and disruption of irrigation for agriculture, water supply and sanitation systems. Over half a million hectares of habitat of conservation importance was estimated to have been affected by the dam breach, from the upstream Kakhovka Reservoir and its wetland habitats to the downstream Black Sea Biosphere Reserve. This impact area covered the distribution of 567 species that have a listing on the IUCN European Red List, 28 of these species have a threat status of vulnerable or worse. There were also concerns about the supply of cooling water to the upstream Zaporizhzhia Nuclear Power Plant, i.e. one war-induced ecological disaster could have resulted in another ecological disaster. Illegal logging, deforestation and charcoal production also support militia in many protracted conflicts throughout Africa (Branch et al., 2023). But, even beyond involvement in war activities, everyday military operations directly generate vast emissions of GHGs (Kester and Sovacool, 2017; Crawford, 2019). The feedback impact of conflicts on the Earth system can also occur indirectly through impeding humanity's ability to collaborate to find solutions to global challenges such as climate change. Within societies entangled in a conflict, resources are diverted to winning the conflict rather than to mitigate climate change, also affecting a country's environmental governance mechanisms. Finally, the continued presence of a large number of tactical and nuclear weapons represents a significant threat to global climate and other Earth system processes (Turco et al., 1983; Xia et al., 2022).

**2.5 Financial Destabilisation**

The impacts of Earth system destabilisation on the financial sector are now receiving increasing attention (Ameli et al., 2023; Chenet, 2024), with studies suggesting that climate-related damages will impact the stability of the global banking system significantly (Lamperti et al., 2019), as can biodiversity loss (Kedward et al., 2023). For instance, stocks of capital at risk due to climate-induced extreme and more frequent weather events such as floods, would adversely affect insurance companies (Lamperti et al., 2019). Reinsurance companies are withdrawing increasingly from areas exposed to high climate change risks, e.g., areas vulnerable to wildfires and floods (Frank, 2023). Earth system destabilisation is likely to result in stranded assets (Caldecott et al., 2021). Escalating climate change can also destroy the capital of firms, reduce their profitability, deteriorate their liquidity, reduce the productivity of their workforce, leading to a higher rate of default, harming the financial sector and the economy in general (Dafermos et al., 2018; Dietz et al., 2021). One issue with the existing empirical evidence and models that try to estimate climate damage for the financial sector is however that they do not account for ESTPs (Keen et al., 2022; Kedward et al., 2023; Trust et al., 2023; Marsden et al., 2024).

Still, first advances are being made. Martin et al. (2024) propose an Integrated Dynamic Environment-Economic model on the coupling of an Earth Model of Intermediate Complexity and a non-linear macroeconomic model in continuous time. Using this model, they found that above a warming of about $+2.3 \circ C$, damages drastically foster the need for additional investments in productive capital for adaptation, which could potentially lead to the emergence of private-debt tipping points and a worldwide cascade of defaults. The inability to repay obligations generates non-performing loans (or bad debt) in the balance

sheets of banks and other financial institutions, with possible systemic implications such as those experienced during the 2008 global financial crisis. It is estimated that climate change will increase the frequency of banking crises by 26% to 248% depending on the extent of climate change (Lamperti et al., 2019). If the banks' equity deterioration due to economic imbalances reaches a certain threshold, secondary systemic effects can be triggered. Financial institutions exposed to troubled banks would suffer losses in the market value of their assets, potentially triggering contagion phenomena (Kiyotaki and Moore, 2002; Yan et al., 2010; Roukny et al., 2013; Chinazzi and Fagiolo, 2015). These contagion phenomena can result in a financial tipping point being reached, when contagion becomes self-perpetuating due to feedback loops in the system that amplify the initial shocks (May et al., 2008; Gai and Kapadia; 2010, Haldane and May, 2011). If ESTPs are triggered, destroying assets and the economic productivity of whole regions, we can expect rapid non-linear tipping effects in the coupled financial sector (Battiston et al., 2017). The financial and economic system would eventually settle into a new state, although this state may be characterized by recession, high unemployment, austerity, and other deteriorating economic conditions. The consequences of such a financial upheaval are often a rapid increase in social instability (i.e. interaction with anomie), increase in radicalisation (i.e. interaction with radicalisation) as more people are forced to compete for basic needs (i.e. interaction with conflict) (Dietz et al., 2021).

This could also impact societies' abilities to mitigate climate change, thus risking the derailment of sustainability transition (Laybourn et al., 2023). Governments will likely try to stabilise financial markets through bailing-out policy such as providing fresh capital and saving insolvent banks and it is predicted that climate change will likely increase the frequency of bailouts (Lamperti et al., 2019). Recent government bailouts in response to COVID-19 have shown a distinct lack of sustainability focus (Rockström et al., 2023). Bailouts negatively affect the public budget and lead to increasing government debts, leaving decreasing resources for addressing Earth system destabilisation, for instance through effective climate change mitigation measures. Financial destabilisation would also deplete businesses and individuals of resources to invest in post-carbon transition (Laybourn et al., 2023).

## 3 Cascading Negative Social Tipping Dynamics

The basis for many tipping point behaviours in social-ecological systems is a non-linear relationship between critical pairs of variables. Non-linearities create disproportionate relationships between cause and effect, potentially leading to change that is faster, more intense, or more extensive than expected (and hence, harder to reverse or control). Cascades, as defined by Klose et al. (2021), are sequential occurrences of events in which an initial event triggers a series of subsequent events and are one important attribute of systemic risk (Sillmann et al., 2022). Cascades are more likely when multiple variables within a given system exhibit and transform non-linear relationships to each other, i.e. when coupled, these relationships transform in ways that often cannot be understood. Crossing multiple negative tipping points in diverse systems increases the likelihood of (partial or localised) societal collapse.

In the context of migration, this can manifest as a domino effect, where an environmental or socio-political event causes involuntary displacement or voluntary migration as people search for improved living conditions and better economic opportunities. This is well documented in the Lake Chad Basin case where climate change and unsustainable resource management affect the sustainability of natural resources, increasing vulnerability and leading to coping strategies such as migration (McLeman et al., 2021). In Ukraine, the war-induced ecological devastation in the aftermath of the Kakhovka Dam destruction has displaced thousands of people, and a major humanitarian programme was initiated in response (WHO, 2023).

A possible tipping cascade can be identified between climate change, food insecurity, and migration. The last five years have seen an increase in food insecurity, representing a problematic reversal of the progress done since the 1990s to reduce world

hunger (FAO et al., 2022). Climate tipping points could dramatically impact food security through direct impacts on production (availability) and indirect impacts on access to food when displacement occurs. One of the most direct ways in which tipping points can affect food insecurity is through changes in rainfall distribution, which would render agricultural livelihoods in rainfed regions unfeasible without irrigation (or other) technologies (Giannini et al., 2017; Benton et al., 2017). Indeed, even in the most optimistic climate mitigation scenarios which would lead to a temporary overshoot over 1.5°C, and then return to temperatures below that threshold, a tipping point might occur in precipitation patterns which can result in adverse food security impacts (cf. Ritchie et al., 2020). Additionally, recent studies suggest that escalating climate change could result in concurrent weather extremes driven by a strongly meandering jet stream, which could trigger simultaneous harvest failures across major crop-producing regions, posing a serious threat to global food security (Kornhuber et al., 2023). Food security can change seasonally. As such, food security does not exhibit traditional bifurcation in the sense of irreversibility. However, a permanent change towards a state of food insecurity would be catastrophic, representing a permanent food crisis. Krishnamurthy et al. (2022) offer a framework to identify "transitions" as prolonged periods of food insecurity (Figure 3), using the Integrated Food Security Phase Classification (IPC), the leading global metric for standardized food security assessment, which combines data on agricultural production, food prices, nutrition rates, weather patterns, and other variables to determine the general food security situation in a given location. With these metrics, a tipping point in a food system can be thought of as a shift between periods with minimal food insecurity (IPC 1 or 2) to periods of sustained food crisis (IPC 3 or higher). An example of a potential tipping point using the IPC categories was found in East Africa in 2015/2016 due to anomalously low rainfall in both the summer and autumn. This trend, combined with insufficient drought preparedness, resulted in crop failures and livestock mortality–and consequently a depletion of livelihood assets, food stocks, and overall food security in northern and eastern regions of Ethiopia (Figure 3).

[FIGURE 3 HERE]

The links between food insecurity and migration are complex, severe food insecurity has been found to trap people locally, who wish to migrate, but are unable to (Sadiddin et al., 2019) but there is also evidence that migration can be driven by food insecurity (Smith and Wesselbaum, 2022). Migration flows are also impacted by climate change directly (i.e. the local environment becomes unsuitable for favourable habitation) and indirectly (i.e. by impacting relative wages through effects on farmers' crop yields). A climate disaster, for instance triggered by a climate tipping point being breached, may also lead to sudden displacement, whether temporary or permanent. To summarise, a cascading dynamic plays out when various tipping points become coupled, for instance, when the tipping in an Earth system triggers the tipping in food insecurity and potentially simultaneously a tipping in displacement, which may in turn reinforce food insecurity.

Other potential cascading links exist as well. For instance, societies may tip into a state of conflict because of competition over dwindling resources as tipping in food insecurity occurs and conflicts in turn may reinforce food insecurity, a cascade made likely when institutions are weak, and governance fails (Martin-Shields and Stojetz, 2019; Anderson et al., 2021; Shemyakina, 2022). Radicalisation and polarisation can fuel conflicts (McNeil-Willson et al., 2019; Rousseau et al., 2021), radicalisation and polarisation has been also observed in countries hosting displaced communities (Ravndal, 2018), a link often moderated by socio-economic inequality and perceived insecurity. Radicalisation, polarisation, and anomie can reinforce each other too. Research suggests for instance that in countries with greater polarisation, people trust each other less (Rapp, 2016). On the other hand, people with mental health issues are more susceptible to conspiracy theories, which can fuel radicalisation (Green et al., 2023). Finally, financial destabilisation can be a driver for radicalisation, polarisation, and anomie (Funke et al., 2016; Bygnes, 2017; Doerr et al., 2022). However, these and other potential cascading links and processes are still little researched and understood.

**4 Emerging research questions and intervention options**

**4.1 Methods and models and emerging data questions**

Various methods and approaches have been suggested for the study of tipping processes in social and socio-ecological systems, which can be used to study negative social tipping points and the cascading interactions between them. In Table 1 we discuss the most prevalent methods and some new emerging approaches. We would like to emphasise here that we are not suggesting that negative social tipping points are knowable in advance, in terms of determining or predicting the exact threshold or time when a tipping will occur. In fact, the knowability of tipping points is a challenge not only for social tipping points but equally for ESTPs (Boulton et al., 2023). It is usually only possible to determine a tipping point subsequently. However, even then there is often not a single negative social tipping point, the exact threshold may vary for instance from one country to another or from one community to another (e.g. c.f. Spaiser et al. 2018 deriving from data specific segregation tipping points for various schools, located on a curve), as the setup of reinforcing and dampening feedbacks will be different in every context. This is also true for some ecological tipping points; e.g. different lakes will have different tipping points (Hessen et al., 2023) The methods we are suggesting here are useful (1) to study tipping processes, once they have occurred or to generate various model-based scenarios to build our general understanding of tipping processes, so we are better equipped to respond to them and (2) to build early warning systems that could potentially capture a system becoming more unstable, chaotic or exhibiting more unusual behaviour before a tipping point has been reached (Dakos et al., 2023). The purpose is to increase our agency (see 4.2).

We are also conscious that all models are oversimplifications of many stories and perspectives and detailed mechanisms. Tipping models can be higher dimensional to capture more dimensions. But even a low dimensional tipping model, such as a neural network (see Table 1), can be used to estimate tipping parameters. In effect a simple model provides a projection of more complex mechanisms in a function space. The main questions are how much information we lose in projecting to a tipping model, compared to a projection to a different model, and how useful the projection is in enhancing our understanding of underlying mechanisms and in determining agency pathways. We believe tipping model development is important to advance our understanding and enhance our agency, but we also advocate the comparison of different models to identify the most useful model.

[TABLE 1 HERE]

Further emerging data questions include:
- What are the most relevant and appropriate datasets for early warning of negative social tipping points? Social tipping points are more complex than physical tipping points due to the interacting relationships between climate parameters and social responses. Given this complexity, there is a need to identify relevant data sources and methods that can be used to detect and anticipate tipping points. Recent advances in machine learning and increasing digital social data all offer an unprecedented opportunity to understand early warning signals for social tipping points. Once datasets are identified, ensuring that these are accessible and usable for analysis is highly important. Moving forward, it will be important to consider sharing platforms to ensure access.
- What are the characteristics of datasets that can render them more (or less) useful for detecting social tipping points? A key, practical question for tipping point analysis is whether there are specific characteristics that make datasets more appropriate for detection of critical transitions. Early warning of tipping points ultimately depends on reliable, high-frequency data (Scheffer et al. 2009, Dakos et al. 2015). For example, in an analysis of data requirements for early warning of food security tipping points, Krishnamurthy et al. (2020) highlighted the importance of temporal resolution over spatial resolution to detect autocorrelation or flickering in coupled climate-food systems. However,

research has shown that even limited datasets such as Soil Moisture Active Passive (SMAP) can provide game-changing opportunities for detecting food security transitions (Krishnamurthy et al., 2022).

- Which early warning signals are more meaningful for different applications? Identifying the most useful metrics and statistics for early warnings of tipping points translates to actionable information, but it requires a clear understanding of underlying system functioning and mechanisms. For instance, in food security applications, autocorrelation is the key metric used to detect a transition in food security states, with the rolling average statistic indicating the direction of the transition (Krishnamurthy et al., 2022). Such insights can help leverage resources in a timely fashion to avert negative effects associated with social systems that exhibit tipping points.

- Moreover, probabilistic insights from research on collective social dynamics may complement insights from new early warning signals for social tipping. These approaches identify measurable qualities of social systems or networks, such as heterogeneity, connectivity and individual-based thresholds that make social tipping points more likely (Bentley et al., 2014). For maximum efficacy, these modelling efforts should derive from both qualitative and quantitative methods so as to benefit from both data and lived experience.

**4.2 Intervention options and emerging policy questions**

Given that negative social tipping points are under-researched, there is little knowledge on how they can be prevented or managed. As noted for instance by Milkoreit et al. (2024), social tipping point governance has not really been developed yet. In Table 2 we nevertheless provide a preliminary overview of potential intervention options, linked to the discussed negative social tipping points and their main potential interactions. Future research needs to focus on identifying other potential intervention options and tying these together into a coherent tipping points governance framework. Ultimately, effective governance of negative (social) tipping points will hinge upon the understanding of collective social dynamics and proactive resource-based interventions. The main line of agency we would like to emphasise is the strengthening of societal institutions and polycentric governance mechanisms (Carlisle and Gruby, 2019; Morrison et al. 2023). We also would like to emphasise agency in driving positive social tipping processes that improve long-term sustainability and well-being of people and planet (Gaupp et al., 2023) and prevent societies sliding into negative social tipping dynamics.

[TABLE 2 HERE]

Further emerging policy questions include:

- How do multiple climate extremes and other shocks and stressors combine, especially do slow onset climate change processes drive systemic changes and tipping points? Evidence provided here, suggests that severe climate events, such as droughts and hurricanes, can result in highly complex social change, including negative social tipping points. Additional research is required to understand if and how climate and social tipping points interact, and whether one tipping point can result in a plethora of other transitions.

- As critical transitions unfold, how does the risk landscape shift in response? Societies respond to environmental stress and resource scarcities. However, these responses may lead to new risks. Understanding how critical transitions affect the current (and future) risk landscape can provide essential information for decision-makers to prioritize investments in adaptation and mitigation.

- What are the processes required to integrate research into policy making? There is growing research on early warning signals for tipping points. However, once suitable datasets and early warning diagnostics are identified, what are the enabling processes and steps required to integrate actionable early warning systems into decision-making? New data analytics, dashboards and communications material may go a long way towards facilitating the transition to early warning systems of tipping points that can translate into action.

**5 Conclusion**

We mapped selected key potential negative social tipping points and their potential cascading interactions. We have also briefly discussed potential intervention options and provided examples of methods and models that need to be advanced in the future. We do not claim to have captured all possible social negative tipping points in the context of Earth system destabilisation, and we acknowledge that other social subsystems could experience negative tipping points as well, e.g. breakdown of (certain) global supply chains (Marcucci et al., 2022), or breakdown of the public health system (at least in certain areas) triggered for instance by a massive freak heat event or the breakout of a disease due to climate change (Sharma 2023, Skinner et al., 2023). Our goal is to highlight that if societies fail to stabilise the Earth system through decarbonisation, land use reallocation and other measures, societies will not merely stay in the business-as usual state. Through mechanisms of negative social tipping accompanying further Earth system destabilisation, they instead risk transitioning into a new social system state, which may be characterised by greater impoverishment, authoritarianism, hostility, discord, violence, conflict, and alienation. Societies more vulnerable to climate change are likely to experience such negative social tipping sooner, but this will inevitably have knock-on effects globally. It is increasingly likely that in some regions large-scale climate adaptation will need to be undertaken to reduce vulnerabilities to the current and future magnitude of climate change.

The acceleration of climate tipping points perpetuates a vicious cycle that weakens societies and their abilities to respond, feeding further Earth system destabilisation. This vicious cycle is also fed by widening socioeconomic inequalities (Millward-Hopkins, 2022). As the consequences of climate change intensify, societal trust, cooperation, and altruism may erode due to increased competition for scarce resources, displacement of populations, and other climate-related challenges. Our knowledge on negative social tipping points is still very patchy and fragmented, with many estimations and models likely to be underestimating the effects of breaching Earth system tipping points. This is particularly true for economic and financial sector models (Marsden et al., 2024). Researchers (Keen et al., 2022) are advocating for developing future loss calculations in close collaboration with climate scientists to ensure adequate representation of climate catastrophes.

**Competing interests**

At least one of the (co-)authors is a member of the editorial board of Earth System Dynamics.

**Disclaimer** For the EU projects the work reflects only the authors' view; the European Commission and their executive agency are not responsible for any use that may be made of the information the work contains.

**Acknowledgements**

Viktoria Spaiser acknowledges support from the UKRI Future Leaders Fellowship award (MR/V021141/1).

Yevgeny Aksenov and Stefanie Rynders acknowledge support from the following projects: COMFORT (grant agreement no. 820989) under the European Union's Horizon 2020 research and innovation program; the EC Horizon Europe project OptimESM "Optimal High Resolution Earth System Models for Exploring Future Climate Changes" under grant 101081193 and UKRI grant 10039429; EPOC, EU grant 101059547; UKRI grant 10038003; and the UK NERC projects LTS-M BIOPOLE (NE/W004933/1), CANARI (NE/W004984/1), and Consequences of Arctic Warming for European Climate and Extreme Weather (ArctiCONNECT, NE/V004875/1). Yevgeny Aksenov and Stefanie Rynders acknowledge the use of the ARCHER UK National Supercomputing and JASMIN.

Weisi Guo acknowledges support from EPSRC Complexity Twin for Resilient Ecosystems (EP/R041725/1).

Jürgen Scheffran and Jana Sillmann acknowledge support under Germany's Excellence Strategy—EXC 2037: "CLICCS—Climate, Climatic Change, and Society"—Project Number: 390683824 funded by Deutsche Forschungsgemeinschaft.

Uche Okpara acknowledges support from the UKRI Future Leaders Fellowship Award (MR/V022318/1)

John T. Bruun gratefully acknowledges the UK Research Councils funded Models2Decisions grant (M2DPP035: EP/P0167741/1), ReCICLE (NE/M004120/1), and STFC Spark Award (ST/V005898/1), which helped fund his involvement with this work.

Graeme S. Cumming was supported by a Western Australia Premier's Science Fellowship awarded through the WA Department of Jobs, Tourism, Science and Innovation.

Jonathan F. Donges acknowledges support from the European Research Council Advanced Grant project ERA (Earth Resilience in the Anthropocene, ERC-2016-ADG-743080) and the European Union's Horizon 2.5 - Climate Energy and Mobility programme under grant agreement No 101081661 (project WorldTrans).

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

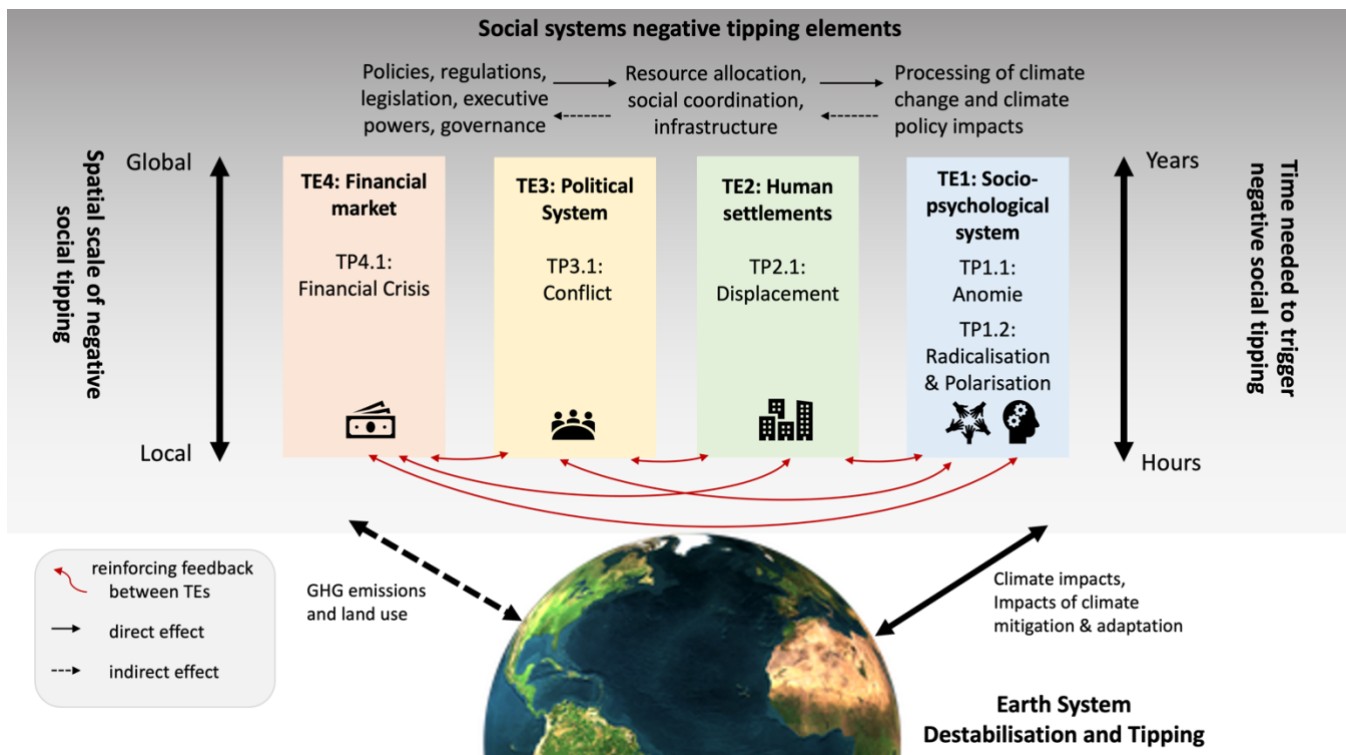


**Figure 1: Tipping elements (TEs) and associated negative social tipping processes (TPs) with the potential to further destabilise the World–Earth system. The identified interactions between the various negative tipping processes mean that they can potentially reinforce one another, making destabilisation more likely. Earth image source: https://pngimg.com/image/25350 (License: Attribution-Non-Commercial 4.0 International (CC BY-NC 4.0))**

1526

**United States**
2018 Camp Fire led to increased chronic psychological issues in the aftermath, such as PTSD and depression

**United Kingdom**
Winter 2013/2014 floods led to increased mental illness. People, whose houses were flooded 50% more likely to suffer anxiety & depression

**Sudan**
Droughts (e.g. in 2015) and desertification led to internal displacement, conflicts and rising mental illness (anxiety, depression, acute stress)

**Vanuatu**
Four natural disasters within one month of 2015 resulted in large-scale destruction, forced displacement and increasing mental illness

**Puerto Rico**
Following Hurricane Maria in September 2017, affected people displaced to Florida showed higher rates of PTSD

**Brazil**
The 2015 Mariana and the 2019 Brumadinho tailing dam catastrophic failures resulted in large-scale destruction and socio-economic loss linked to mental illness and suicide.

**South Africa**
Between 2008 and 2022 higher rates of substance misuse have been reported among people displaced due to climate change or exposed to extreme climate stressors

**India**
Increasing suicide rates among rural farmers in India during climate-induced heat waves (e.g. in 2022) associated with crop failures

**Thailand**
The 2011 flooding in Thailand put victims at 50% higher risk of serious mental illness (higher levels of anxiety and depression)

**Australia**
Prolonged droughts (e.g. 2017-2019) disrupted pastoral farming, forcing Indigenous Australians to migrate to town, which has negatively impacted their mental health

**Figure 2: Examples of the impact of extreme weather events on mental health across the world, based on Carleton (2017); Clayton et al. (2017); Jermacane et al. (2018); Atwoli et al. (2022); Hamideh et al. (2022); Lawrence et al. (2021), and Ferreira et al. (2023).**

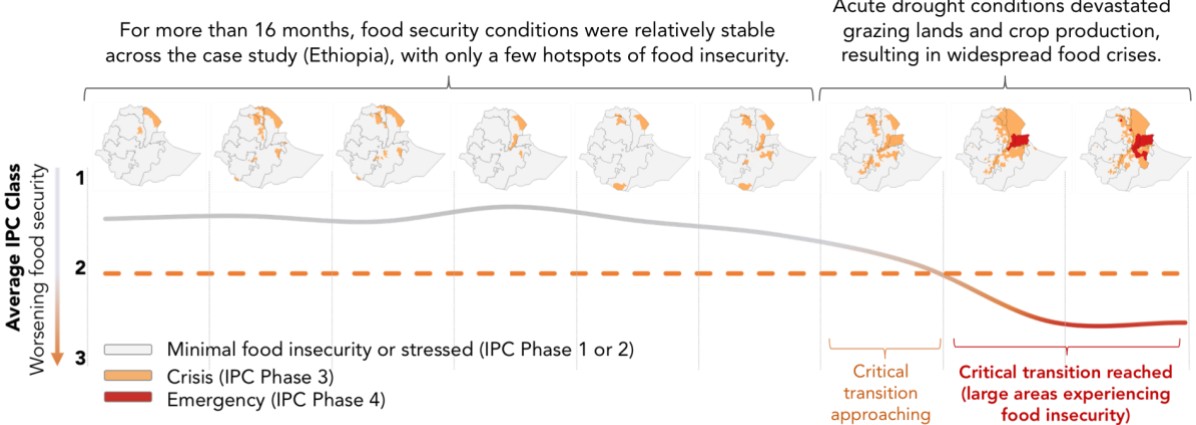

**Figure 3. Example of a "tipping point" in the context of food security, showing the transition from stable food security conditions to a food crisis resulting from drought in Ethiopia (Source: Krishnamurthy et al., 2020)**

**Table 1. Models and Methodological Approaches for Studying Negative Social Tipping Points and Cascades**

| Model/ Approach | Rationale | Modelled phenomena | Examples | Further Questions |
|---|---|---|---|---|
| (Complex) Contagion Processes on (Social) Networks | In a simple contagion direct exposure to a viral entity (beliefs, behaviours, emotions, price signals) is sufficient for a node to get "infected". In a complex contagion a node gets "infected" if a certain number (can be heterogeneous) of its neighbour nodes are infected (Guilbeault et al., 2018; Wiedermann et al., 2020; Andreoni et al., 2021). Models of contagion on networks can be used to study radicalisation, anomie, and financial tipping. | In a contagion a viral entity spreads initially gradually until a critical threshold (critical number of "infected" nodes) is reached at which stage the social system tips through saddle-node bifurcations and hysteresis. Hysteresis ensures that the contagion spreads further and leads to the phase transition, even if the original seeders of the viral entity are removed from the network, i.e., the contagion processes become self-reinforcing (Dodds and Watts, 2004; Wiedermann et al., 2020; Xie et al., 2021). Network structure (e.g. clustering) can facilitate or prevent various contagion processes (Guilbeault and Centola, 2021). | Research shows that beliefs (incl. misinformation), mental states, behaviours and practices (e.g. technology adoption) can spread through complex contagion across social networks (Karsai et al., 2014; Törnberg, 2018; Fink et al., 2021; Xie et al., 2021; Alexander et al., 2022). Research on financial contagion also shows that volatility can spread across a network of financial institutions (Summer, 2013; Wunderling et al., 2021). | There are gaps in our understanding of the mechanisms underlying complex contagion in the real world, where at any given time multiple, conflicting diffusion processes are taking place (Min and Miguel, 2018; Vasconcelos et al., 2019; Yletyinen et al., 2021). |
| Logistic Maps Models | The logistic map is a mathematical function that models the population change of an ecosystem over time and it is a useful tool for policy and climate analysis as it represents a wide range of regular and chaotic features (Feigenbaum, 1980; Bruun et al., 2017). Logistic Maps can be used to study anomie social tipping and cascading dynamics for instance in financial and political systems (incl. conflicts). | The logistic map provides the capability to investigate non-abrupt and/or reversible tipping point changes that are features of the system. It represents the socio-economic system through the population level, at time t, as $X_t$, and its future population state at time t+1 is specified by the non-linear relationship $X_{(t+1)} = r X_t (1-X_t)$. It enables us to identify and explore tipping point transitions and complexity cascades properties across a set of different system types. | Logistic maps have been used to model financial and economic cycles and crises (Ausloos and Dirickx, 2006; Guégan, 2009). Logistic maps have also been employed to study conflicts (Guastello, 2008; Scheffran and Hannon, 2007). | The model could be useful to study phenomena such as anomie, where the ecological and social system are closely coupled and the tipping in the ecological system would have direct repercussions for the social system with one possible outcome being disintegration of the social system, i.e. chaotic, random and irregular behaviour of the social system. |
| Causal Loop Diagrams (CLD) and Causal Inference | Causal loop diagrams (CLD) are a structural approach for systemic risk assessment on different scales and to identify whether a society is at risk of reaching a negative social tipping point (Groundstroem and Juhola, 2021; Sillmann et al., 2022). Causal inference is the attempt to empirically test causal assumptions. CLDs and causal inference can be used to study displacement, conflicts, and cascading dynamics. | CLDs map out the structure of a system and its networks and reveal causalities and feedbacks within the system (Haraldson, 2004; Sanches-Pereira and Gómez, 2015). Variables are connected with arrows that indicate positive or negative causal links between them. Links between variables may have temporal delays (Sanches-Pereira and Gómez, 2015). Feedback effects arise when variables affect each other in a cascading manner, ultimately leading back to a previous variable, creating a feedback loop. This loop can be either reinforcing (R), leading to unbounded growth or decline, or balancing (B), if some variables create counteracting changes, resulting in equilibrium. | CLDs have been used to model socio-ecological system dynamics, for instance the coupling of climate change, food insecurity and societal collapse (Richards et al., 2021). Causal inference has been used to model for instance climate induced conflict as an excitation causal process (Sun et al., 2022). Machine learning methods have also been used for causal inference, i.e. to self-discover causal trees between Earth system and social systems, including climate conflict (Ge et al., 2022). | Improving causal understanding of how changes in the Earth system affect social systems is challenging when many of the latent mechanisms and pathways lack data, and when different regions experience diverse mechanisms. End-to-end causal inference has limited success (Guo et al., 2023). |

| | | | |
|---|---|---|---|
| Multi-Stable Differential Equation Models | Approaches building on mathematical dynamical systems theory (Hirsch et al., 2012), analyse time series data to identify possible phase transitions from stability to instability until a new equilibrium is found. Differential equation models can have multiple equilibrium points, where the rate of change of a variable (e.g. degree of cooperation) does not change further. These models can be used to study negative social tipping phenomena, where sufficient time-series data is available, e.g. conflicts and financial systems tipping. | The specific functional form of the models can vary depending on the studied phenomenon. A tipping model can be for instance a $3^{rd}$-order polynomial in the form of a bi-stable ordinary differential equation (ODE): $dx/dt = x(x-C)(x-K)$. Here, we can see that the rate of change ($dx/dt = 0$) has three equilibrium points: $x=0$, $x=C$, $x=K$. Two of the three equilibria are stable, i.e. a small perturbation will cause the system to return to the closest point 0 (conflict) or $K>C$ (cooperation). One of the three is unstable, i.e. a small perturbation will cause the system to deviate away completely (this is the tipping criticality point C). | Multi-Stable Differential Equation Models have been also used for assessing the risks of emerging tipping cascades in interconnected climate tipping elements (Krönke et al., 2020; Wunderling et al., 2023) and financial systems (Wunderling et al., 2021) using Monte Carlo approaches to propagate parametric and structural uncertainties. They have also been used to study conflict dynamics (Aquino et al., 2019). | The models rely on rich and dense multiple time-series data. They are also constrained in terms of complexity representation. This results partly from their aggregate nature, as they are mainly concerned with macro-level dynamics; as such they might be less suitable where micro-level interactions are of interest. |
| Agent-Based Modelling (ABM) | Agent-Based Modelling (ABM) represents the rule-based behaviour and interaction of individual agents which ranges from simple homogenous to complex heterogeneous agents characterised by diverse response functions regarding their motivation and reasoning, capability to act and adaptive learning, perception, and anticipation of changing environmental situations (BenDor and Scheffran, 2019). ABM can be used to study conflicts and cascading dynamics. | Multiple agents show collective behaviour via opinion dynamics, coalition formation, network building, inducing social feedback, structural shifts, social norms, and transformative policies, including the transition between conflict and cooperation (Juhola et al., 2022). ABM captures macro-scale phenomena from micro-scale interactions among many heterogeneous adaptive and learning agents with bounded rationality (Filatova et al., 2013; Weber et al., 2023). | ABM is applied to study agents' adaptation behaviour and the possible limits to adaptation (Juhola et al., 2022). ABM approaches are well suited to model game-theoretical approaches to predict agent-induced tipping points when collaboration for instance breaks down (Grimm and Schneider, 2011). They can simulate self-reinforcing chain reactions and cascading effects in dynamic social networks (BenDor and Scheffran, 2019). | Where ABM lacks empirical foundation (i.e. insufficient data for a large number of agents), it is difficult to verify the predictions they are making. They can be useful to generate hypotheses and explore theoretical mechanisms, which should be tested empirically. |
| Machine Learning (ML)/AI | Machine Learning (ML) approaches have been already mentioned in the context of previous sections (causal inference). But ML methods can also be used to explicitly detect tipping points (Bury et al., 2021). ML can be used to study negative social tipping phenomena, where sufficient time-series data is available, e.g. conflict, financial systems tipping etc. (Ge et al., 2022). Generative AI is also discussed for the purpose of generating in-silico data (fine-tuned by human data) (Argyle et al., 2023; Park et al., 2023; Törnberg et al., 2023), e.g. high-dimensional, dynamic social network data for in-silico large-scale experiments, mimicking real life and real people, to study otherwise difficult to study phenomena, such as negative social tipping processes. | ML models have been used for instance to model bifurcations, i.e. the divergence of an outcome trajectory. These are often mechanism-informed ML models. Hawkes excitation model has been used for instance to model the coupling between successive improvised explosive device (IED) attacks and security retaliation (Tench et al., 2016). Point process modelling has been used to identify complex underlying processes in conflicts, such as diffusion, relocation, heterogeneous escalation, and volatility (Zammit-Mangion et al., 2012). | ML approaches can be useful to forecast tipping in conflicts for instance (Guo et al., 2018). With increasing availability of rich digital data, negative social tipping processes (e.g. radicalisation or social disintegration) could be detected using for instance Deep Learning models in combination with social network analyses (Gaikwad et al., 2022). ML-based tools are also emerging to predict tipping in financial systems (Samitas et al., 2020) | Pure data driven prediction models (e.g. using Gaussian Processes, Deep Recurrent Neural Networks), typically lack the ability to model irreversible transformations, such as tipping and understand causal relation strength. But if sufficient data is available and if the ML models are informed by theory and deep understanding of the underlying mechanisms (Guo et al., 2018) they can be a useful method. |

1541 **Table 2 Negative (social) tipping points and options for prevention and impact management**

| Negative (Social) Tipping Points | Prevention Options | Impact Management Options |
|---|---|---|
| Earth System Tipping Impacts (e.g. food insecurity) | Early warning systems to detect escalating food insecurity and anticipatory action mechanisms, incl. investment in irrigation, crop diversification and investment in long-term adaptation options to improve climate-smart agriculture (Krishnamurthy et al., 2020) | Risk finance (e.g., weather index insurance) (Benso et al., 2023) and emergency response (e.g., food assistance), managed relocation from areas that become uninhabitable/uncultivable (Ferris and Weerasinghe, 2020). |
| Anomie | Strengthening resilience of individuals and communities (Ogunbode et al., 2022). Strengthening social cohesion (Orazani et al., 2023). Ensuring authorities can respond to ecological hazard effectively through capacity building and resilient infrastructure (Miller, 2016; Brown, 2020) | Mental health support to individuals and communities affected by extreme weather events and displacement (Wood and Kallestrup, 2021). Working with affected communities to re-build and integrate displaced communities in host communities (Hawkins and Maurer, 2011) |
| Radicalisation & Polarisation | Preventing the spread of misinformation/ disinformation (Aïmeur et al., 2023). Psychological inoculation against misinformation/disinformation (Van der Linden et al., 2017). Monitoring radicalisation. Radicalisation prevention programmes. Public engagement in democratic, deliberative decision making (Devaney et al., 2020). | Deradicalization and dialogue building programmes (Kimmel, 2018; Hangartner et al., 2021). Containing the influence of radical groups (Flache et al., 2017). Early warning systems for detecting the potential for violence (Guo et al., 2018). |
| Displacement | Early warning systems and anticipatory action mechanisms, e.g. managed relocation. Investing in resilience of displaced communities, through stability, education, and employment opportunities (Ferris and Weerasinghe, 2020). | Host community and refugee support (e.g., humanitarian support, food aid, housing, mental health support) (Pearce et al., 2017). Financial compensation for host communities. Legal frameworks and policies to support mixed movements (McAdam, 2012) |
| Conflict | Conflict early warning systems (CEWS) (Guo et al., 2018). Conflict prevention processes, through conflict management and democratic procedures. Agreements on scarce resource management and distribution. Climate change adaptation support. Resilience building of societies at risk of violent conflict (Abrahams, 2020). Conduct conflict risk assessment of critical infrastructure identifying impact cascades across rural, urban and natural environments to inform redevelopment or security measures to mitigate risks. | Conflict resolution process (Ngaruiya and Scheffran, 2016). Humanitarian support to citizens trapped in conflicts. Managed relocation from active fighting zones. Provision of evidence to support post-conflict reconstruction and recovery building. Provision of clean water, sanitation, hospitals, and schools. Biodiversity recovery planning to restore critical habitats and species, including those of high economic value to support social recovery. |
| Financial Destabilisation | Early and stable transition away from fossil fuel assets (i.e. divestment). Implementation of a green corporate quantitative easing programme to reduce climate-induced financial instability and restrict global warming (Lamperti et al., 2019) | Macroprudential regulation in climate risk management. A counter-cyclical capital buffer (as proposed in the Basel III framework) could help address climate physical risks, even though it may be insufficient when damages surge (Lamperti et al., 2019) |
| Cascading dynamics | A big potential lies in recovery and reconstruction efforts that have the goal to build resilience to prevent future negative social tipping points cascading (Hanson, 2018). During recovery and reconstruction planning, options for climate change adaptation and biodiversity recovery may provide a level of risk management for future conflict and natural disasters (e.g. adapting to future flood risk in the lower Dnipro basin caused by climate change in Ukraine). An approach to assessing impact cascades may be transferable to risk assessment and mitigation of natural disasters globally (Ward et al., 2020). The United Nations Disaster Risk Reduction (UNDRR) Programme Framework may provide a starting point for such an approach (UNDRR & ISC 2020). | Overall, management options for cascading impacts have been studied relatively little. Management options depend greatly on the type of cascading impact and the systems between which it occurs. In general, collaborative governance, bilaterally or multilaterally, between governing entities can yield better outcomes. |

1542