# Peer review of "Negative Social Tipping Dynamics Resulting from and Reinforcing 1 Earth System Destabilisation 2"

_EGUsphere, 2023_

## Author Response (AR1)

**Response to Reviewers**

We thank both reviewers for their constructive and helpful feedback, which helped us a lot to refine our paper.

**Reviewer 1**

We are glad to hear reviewer 1 enjoyed reading the paper.
We have provided an extensive response to the comments in the Discussion section of the original manuscript. Here we will briefly summarise the changes we made in the revised manuscript in response to the comments. All changes can be tracked in the tracked changes manuscript.

*Abstract: The abstract could profit from naming the identified negative social tipping dynamics.*

We have revised the abstract, including naming the identified negative social tipping dynamics.

*L 116: It is unclear how a study ENSO relates to perceptions and anomie... is the Bruun et al., 2017 citation correct in the reference list?*

We have added a brief explanation and a refence to Table 1, where logistic maps are explained in greater detail. Bruun et al. 2017 is used to reference the methodology, which is widely applicable, rather than specifically ENSO.

*L142: "When climate is ..." should read "When climate change is ..."*

Changed.

*L145: "..., which can include deliberate polarisation of societies on the issue". Can you briefly elaborate what kind of polarization is meant here? ...*

We have revised and expanded this section significantly; the changes should make it clear what kind of polarization we are discussing.

*L183-186: You are referring to a complex systems lens and the HECS/SFL framework. (...), could you briefly concretize how a complex systems lens would be  helpful...*

We have expanded the explanation on this and made links between the HECS and SFL frameworks more explicit.

*L196 – Comment on section "Financial Destabilisation": I liked to read this section but I expected to see at least a brief discussion on the economics of crossing climate tipping points ... In particular, I was thinking of literature along the following lines (These are mostly IAM-based results): https://www.pnas.org/doi/10.1073/pnas.2103081118*

We have included the suggested reference and mention briefly general economic implications of tipping points throughout the paper as well as within the section on Financial Destabilisation.

*L248-250: (...) Even in the "most"-optimistic emission scenarios that lead to a temporary overshoot over 1.5°C AND then return to temperatures below, can lead to regionally different precipitation patterns (10 NICS, Insight#1, doi:10.1017/sus.2023.25; original reference: Kim et al., 2022: https://doi.org/10.1038/s41558-022-01452-z), which may endanger stable food production. I also think that the following paper would be relevant to mention, which is one of the very few studies of how an Earth system tipping point may impact the food system: UK food system after AMOC tipping point: https://www.nature.com/articles/s43016-019-0011-3*

We have added some explanation on this and included further references, highlighting the dangers of overshoot, but also highlighting how food insecurity can be underestimated by not accounting for simultaneous harvest failures.

*Fig. 1: I am unsure whether the two axes on the left (spatial scale) and right (Time needed to trigger) are helpful because there is no order among the four major TEs; only a comparison between TP 1.1 and 1.2 can be made. I recommend to remove the axes or more clearly map the four TEs according to time and space.*

We have discussed this in the author team but then decided not to remove the axes as they are important in order to highlight that tipping can occur at various temporal and spatial scales. We have added a sentence explaining this in the text now, which hopefully makes it easier to read the figure.

*Fig. 2: it would be helpful to add at least a rough time frame to those boxes where this is not already mentioned to unify the notion of the boxes (e.g. for Sudan or India, rough times are missing).*

Time frames were added in the figure now for the boxes where they were missing in the previous version of the figure.

*Fig. 3: Please add a horizontal time axis with the years, and maybe an additional marker where strong El-Niño phases occurred (if the authors think that this may have been a trigger of food insecurity in Ethiopia; if the authors don't think El-Niño played a large role, I recommend to remove it from the main text) so that the figure can be more easily understood.*

This is a published figure that has been republished in the manuscript, we are hence unable to change it unfortunately.

**Reviewer 2**

We are thanking reviewer 2 for a critical assessment of our paper.
We have provided an extensive response to the comments in the Discussion section of the original manuscript. Here we will briefly summarise the changes we made in the revised manuscript in response to the comments. All changes can be tracked in the tracked changes manuscript.

*I'm not exactly sure how to evaluate the piece as the piece is pretty abstract, short, and references a host of other literature. (…). I think the piece is trying to generate a research program in this space by posing questions, suggesting answers and methodologies, but I think it might want to revisit the accepted premise that there are social tipping points that are knowable. (…)*

Our revised manuscript contains now several brief case studies and more extended explanations, which hopefully will make it less abstract and more useful as a piece to inspire future research. We are also discussing more explicitly and critically the knowability of social tipping points in section 4.1.

*My main concern is that I'm not sure how successful the search for social tipping points is likely to be. I'm not sure how operationalizable these phenomena are. Even if social tipping points are real (and I'm not convinced they are), they may be fundamentally unknowable. While I think there are negative feedback loops between physical and social systems, the effort to specify when tipping points have been breached may be something of an unsatisfying search for mathematical precision to depict relationships that can't be precisely specified that way, given human agency.*

We have provided an extensive reply to this comment in our reply, here we would like to point out that we have extended our discussion and explanation of what we mean by negative social tipping points and their links to Earth system and ESTPs in the introduction, throughout the discussion of various tipping processes and in 4.1. We are also more explicitly discussing now the implications for and role of human agency, e.g. in the introduction and 4.2.

*I think it's not quite clear at what unit of analysis social equilibria purportedly exist – are these properties that exist at the level of the international system, states, or communities (or all of them).*

We have added a sentence in reference to Figure 1 within the main text to explain that tipping can occur at various temporal and spatial scales.

*Concepts such as anomie, which is a nod to Durkheim's classic 19th formulation of a breakdown in social order, seem to be especially fuzzy and difficult to attribute back to climate change. (…)*

We have thoroughly revised the section on anomie to aid clarity on the concept, how tipping dynamics can unfold that are linked to the concept and how these processes are linked to the Earth system. An added case study is also meant to make it less abstract.

*Line 117 references one way to understand social tipping points with respect to anomie is the use of Logistic Map models which I've never heard of. They are described in a bit more detail in Table 1. It would be useful to note that the Models are described in more detail in the Table. An example of a specific geographic application would help.*

We have added a brief explanation as well as a reference to Table 1. A specific case study has been added on anomie (not explicitly linked to logistic maps). Also, examples of applications of logistic maps to study social phenomena have been added in Table 1.

*The piece is most convincing when it uses an example like the example of food insecurity depicted in Figure 3 of Ethiopia and discussed in detail on page 7. The promise of work like this is most convincing where there are concrete examples.*

Thank you. We have now included concrete examples in each section.

*There is a tendency for the introduction of technical jargon and verbiage which is underexplained and inaccessible unless you are already familiar with the studies and approach. For example, line 300 on page 8 about autocorrelation and food security is not intelligible unless you already know what it is referencing. Similarly, line 305 about heterogeneity and connectivity don't make much sense on their own.*

We have discussed in the author team how to respond to this comment. We have decided not to include explanations for terms (such as autocorrelation), where we believe they are broadly understood within the community that this paper is addressing. Adding explanation can disrupt the flow of the text and for most readers they will not be necessary. Less common terms are explained (e.g. in Table 1).

*Line 125 – derogative – (not sure this is a word) – derogatory?*

Corrected